# Research on the Mechanical Properties of EPS Lightweight Soil Mixed with Fly Ash

**DOI:** 10.3390/polym16243517

**Published:** 2024-12-18

**Authors:** Lifang Mei, Yiwen Huang, Dali Xiang

**Affiliations:** 1School of Civil Engineering, Architectural and Environment, Hubei University of Technology, Wuhan 430068, China; meilfhg@163.com (L.M.); 102110966@hbut.edu.cn (D.X.); 2Key Laboratory of Intelligent Health Perception and Ecological Restoration of Rivers and Lakes, Ministry of Education, Wuhan 430068, China

**Keywords:** fly ash, EPS beads, lightweight soil, mechanical properties, micro

## Abstract

Expanded polystyrene (EPS) bead–lightweight soil composites are a new type of artificial geotechnical material with low density and high strength. We applied EPS bead–lightweight soil in this project, replacing partial cement with fly ash to reduce construction costs. EPS beads were used as a lightweight material and cement and fly ash as curing agents in the raw soil were used to make EPS lightweight soil mixed with fly ash. The EPS bead proportions were 0.5%, 1%, 1.5%, and 2%; the total curing agent contents were 10%, 15%, 20%, and 25%; and the proportions of fly ash replacing cement were 0%, 15%, 30%, 45%, and 60%, respectively. Unconfined compressive strength (UCS) and scanning electron microscopy (SEM) tests were conducted. The results showed that the EPS content, total curing agent content, and proportion of fly ash replacing cement had a significant impact on the UCS of the lightweight soil. This decreased with an increase in EPS content and decrease in total curing agent content and decreased with increased proportions of fly ash replacing cement. When the proportion of fly ash replacing cement was not too high, the strength of the lightweight soil decreased less, and its performance still met engineering needs. At the same time, the soil can also consume fly ash and reduce environmental pollution. EPS lightweight soil mixed with fly ash still has advantages, and it is recommended to keep the proportion of fly ash replacing cement less than 30%. The failure patterns for lightweight soil mainly include splitting failure, oblique shear failure, and bulging failure, which are related to the material mix ratio.

## 1. Introduction

With the construction of various high-rise buildings, express highways, and cross-river and cross-sea bridges come more and more problems relating to geotechnical engineering, such as slope instability, bridgehead jumping, the cracking of underground pipes, etc. The main reason is the great weight of the upper filling soil. Expanded polystyrene (EPS) is a kind of polymer material with stable chemical properties. It is difficult to degrade after use, making it prone to causing white pollution [1,2,3,4,5]. It is crucial to recycle expanded polystyrene [6,7,8,9,10]. Lightweight soil made by mixing EPS beads has the characteristics of lightness and high strength, which can effectively reduce soil weight, consume waste EPS, and reduce pollution. As such, research on EPS lightweight soil (ELS) is of great significance.

In 1965, Norway first applied lightweight soil in road engineering, successfully addressing the hazard posed by vehicle jumping at bridgeheads. Subsequently, more and more countries conducted research on lightweight soil. Lightweight soil has been widely used in geotechnical engineering [11,12,13,14,15,16,17,18,19], and many scholars have conducted research on this topic. Mei [20] investigated the physical and mechanical properties of ELS throughout freeze–thaw cycles. EPS beads and cement effectively reduced the frost heave rate, mass loss rate, and compressive strength loss rate of ELS, but the performance of ELS decreased when the content of EPS beads was more than 2%. Shirazi et al. [21] studied the shear and compression behaviors of sandy and clayey soils mixed with EPS using direct shear and unconfined compressive strength (UCS) tests. The internal friction angle, dilation in the sand, and the UCS of the clay–EPS samples decreased with an increase in EPS content. In contrast, an increase in apparent cohesion in the sand–EPS samples was observed. Tiwari et al. [22] added EPS beads to expansive soil and found that when the EPS bead content was not more than 1%, the swelling pressure and expansion percentage decreased and the stability and safety factor of the expansive soil road base increased. Silveira et al. [23] added EPS beads to clayey soil, bentonite, and sandy soil. For the clayey soil, the EPS beads decreased the maximum dry density and optimum water content of the material; for bentonite, with the increase in EPS bead content, the stiffness decreased and residual resistance increased; and for sandy soil, the EPS beads affected its stress–strain behavior when confining pressure was greater, while for lower confining pressure, the EPS beads did not affect shearing resistance. Hou [24] studied the durability of ELS. The effect of dry–wet cycles under soaking conditions on the mechanical properties of ELS were researched via density tests, water absorption tests, and UCS tests. Under soaking conditions, the physical properties of ELS ground were extremely stable. When the soaking time was more than 90 days, the UCS of the specimens reached a stable value. Under dry–wet cycles, the ELS did not appear to crack, and maintained its density and mechanical properties. Thus, ELS has good durability and water stability. Mei [25] carried out dynamic tests on ELS. The dynamic strain and damping ratio increased with the increase in EPS content under the same dynamic stress conditions and decreased with the increase in cement content and confining stresses, but dynamic shear moduli changed in the opposite direction.

The partial replacement of traditional curing agents with environmentally friendly materials has recently become a commonly used method. Zeolite is a pozzolanic material that can replace partial cement due to its superior environmentally friendly properties. Khajeh [26] investigated the stress–strain behavior, peak strain energy, UCS, and CBR of zeolite- and cement-treated sand–EPS bead mixtures. Zeolite (at an optimum replacement proportion of 30%) improved the UCS and CBR values significantly, and the reduction caused by EPS beads was well compensated. Khajeh [27] also conducted systematic surveys on the changes in mechanical strength and shear stiffness of untreated and lime–zeolite-treated clays induced by the incorporation of EPS beads. The optimum content of zeolite used to replace lime was 25%, yielding the highest UCS. Incorporating EPS beads decreased the samples’ swelling and compaction properties, and the strength and stiffness characteristics of ELS were acceptable when the EPS bead content in the mixtures was 0.1%.

The common curing agent used in ELS is cement, but the production of cement causes serious environmental problems. At present, the amount of coal being combusted is increasing, producing a large amount of fly ash. Improper treatment of fly ash pollutes the environment and jeopardizes human health [28,29,30]. Fly ash has been used as a curing agent in concrete [31,32,33,34,35] and can improve the properties of concrete and reduce the cost of materials. Fly ash is also widely used in geotechnical engineering [36,37] and can not only ensure the strength of soil, but can also protect the environment and reduce pollution. Turan et al. [38] found that incorporating fly ash increased the plastic limit, UCS, optimum moisture content (OMC), cohesion (C), and angle of internal friction (φ) of soil and decreased the liquid limit and maximum dry density. They also found that C-class fly ash was more effective than F class. Hu et al. [39] improved granite residual soil by adding fly ash, whereafter the permeability coefficient of the residual granite soil reduced when the fly ash content was 15%; the triaxial strength of improved soil increased most significantly, and the stability of the soil slope was significantly improved compared to the original slope. Li et al. [40] found that adding fly ash to cement soil can reduce its vertical compression deformation and compression coefficient and improve its resistance to deformation and compression. Cheng et al. [41] found that freezing and thawing cycles reduced the UCS, triaxial shear strength, cohesion (C), and angle of internal friction (φ) of salinized soil, while the addition of appropriate amounts of fly ash improved the UCS, shear strength, cohesion (C), angle of internal friction (φ), and performance of freeze–thaw cycles of salinized soil, with 15% fly ash being the optimal content.

Previous research on ELS mainly focused on soil mixed with cement. Fly ash is commonly used in construction, and it is a high-value-added filler material. Replacing cement with an appropriate amount of fly ash as a curing agent can reduce the amount of cement used, lower costs, and facilitate economic and environmental protection. At the same time, it can consume waste fly ash, allow the large-scale utilization of fly ash, and accelerate the transformation and upgrading of waste utilization. There is a large amount of silty clay in plain areas. In this case, in order to render ELS more widely applicable, facilitate the use of local materials, and lower costs, it is necessary to use silty clay as raw soil. Mixing fly ash and cement as curing agents, EPS beads as a lightweight material, and silty clay as raw soil leads to the formation of EPS lightweight soil mixed with fly ash. There are few research reports on this type of lightweight soil, currently constituting a new direction of research. Due to its lightness and high strength, EPS lightweight soil mixed with fly ash has broad application prospects in areas such as soft foundation treatment, highway widening, pipeline landfilling, bridge abutment filling, and retaining wall soil backfilling. Through UCS and SEM tests, we carried out a study on the physical and mechanical properties and microstructure of EPS lightweight soil mixed with fly ash to provide a theoretical basis and reference for future related engineering and research.

## 2. Materials and Methods

### 2.1. Test Materials

The raw soil used in this test was silty clay from the foundation pit of a construction site in Wuhan, where the depth of the soil was 5 m underground. The raw soil is shown in Figure 1, and its physical parameters are shown in Table 1. The lightweight material used was spherical EPS beads with a diameter of 1–2 mm. The EPS beads came from a plastic factory and had a pure particle density of 0.034 g/cm^3^ and a bulk density of 0.023 g/cm^3^. The EPS beads and a grain-size curve are shown in Figure 2 and Figure 3. The curing agents were ordinary 42.5-grade Portland cement and high-quality first-grade fly ash; their physical parameters are shown in Table 2. The grain-size curve of fly ash is shown in Figure 4. Before the experiment, the raw soil was dried in an oven at 105 °C. The dried soil was crushed and sieved to 1 mm, and the screened dry soil was stored in a sealed bag for future use in this experiment.

### 2.2. Specimen Preparation and Test Scheme

The content is the proportion of the mass of each raw material to the mass of dry soil (based on the mass ratio standard). The EPS bead contents were 0.5%, 1%, 1.5%, and 2%; the total curing agent contents were 10%, 15%, 20%, and 25%; the water content was optimum; and the proportions of fly ash replacing cement were 0%, 15%, 30%, 45%, and 60%, respectively. The proportions of fly ash used to replace cement were added by weight. The test scheme is shown in Table 3.

The specimens used in the UCS test were cylindrical specimens with a diameter of 39.1 mm and a height of 80 mm. According to the set ratio, dry soil, cement, and fly ash were mixed evenly, fully mixed with an optimal amount of water, and stirred into a homogeneous mixed slurry; then, EPS beads were added and stirred for 10 min to form a homogeneous mixture of EPS bead lightweight soil mixed with fly ash. Subsequently, a layer of filter paper was placed on the bottom of each of the three petal molds, and the inner wall was evenly coated with petroleum jelly. The lightweight soil was divided into 3 layers—with each layer being compacted 27 times—and loaded into the three petal molds (with a diameter of 39.1 mm and a height of 80 mm), and each layer was made to a depth of about 1 cm until the formation of a dense specimen. Finally, the prepared specimens and molds were placed in a standard curing box with a curing temperature of 20 ± 2 °C and a relative humidity level greater than 95%. After 24 h of curing, the molds were removed, and the specimens continued curing until 28 d had passed. Three parallel specimens were prepared for each group of tests, and the average value was taken to be the final experimental result.

The strain rate was set to 1 mm/min during the UCS test. The specimens used for the SEM test are the same as those used for the UCS test. Three small squares were taken from each specimen and coated for the scanning electron microscopy test.

## 3. Results and Discussion

### 3.1. The Effect of EPS Content on the Density of Lightweight Soil

EPS beads are lightweight materials with low density, good seismicity, stable chemical properties, low cost, etc. Mixing EPS beads into raw soil can reduce the self-weight of the soil and soil pressure. To study the effect of EPS beads and a curing agent on the density of lightweight soil, 16 groups of specimens were selected for the test, with a proportion of cement replaced with fly ash equal to 30%; 10%, 15%, 20%, and 25% total curing agent contents, and 0.5%, 1%, 1.5%, and 2% EPS content. The density of the specimens after curing for 28 days is shown in Figure 5.

When the total curing agent content was unchanged, the density of lightweight soil decreased significantly with the increase in EPS content. As the EPS content increased by 1%, the density decreased by at least 10%. With a 10% total curing agent content and a 0.5% EPS content, the density was 1.45 g/cm^3^. When the EPS content increased to 1%, 1.5%, and 2%, the density decreased to 1.34 g/cm^3^, 1.25 g/cm^3^, and 1.12 g/cm^3^, decreasing by 7.6%, 13.8%, and 22.8%, respectively. When the EPS content was unchanged, with the increase in total curing agent content, the density increased slightly, with a difference of 5% regarding the total curing agent content, while the difference in density was only 0.01–0.03 g/cm^3^. The effect of the curing agent on the density of lightweight soil is small. Thus, it can be gleaned that the EPS content is the main factor affecting the density of lightweight soil; with an increase in EPS content, density gradually decreases. Therefore, EPS beads can be used to reduce the weight of soil.

### 3.2. Stress–Strain Characteristics of Raw Soil

To facilitate a comparison with lightweight soil, we also conducted a UCS test on raw soil. Due to the absence of curing agents, the UCS of raw soil is relatively low. The unconfined stress–strain behavior is shown in Figure 6, and the maximum stress is 198 kPa. After the peak stress, the stress rapidly decreases, showing strain softening.

### 3.3. Effect of EPS Content on Stress–Strain Curves

Four groups of specimens were selected, featuring a 10% total curing agent content, a 0% proportion of cement replaced with fly ash, and 0.5%, 1%, 1.5%, and 2% EPS bead contents; the corresponding stress–strain curves are shown in Figure 7. With the increase in EPS content, the peak stress of the specimen decreased, destructive strain increased, and the UCS of the lightweight soil decreased. When the EPS bead content increased from 0.5% to 2%, the UCS of the lightweight soil decreased from 1410 kPa to 730 kPa, and its strength decreased by 48.2%. The stress–strain curve was a straight line in the initial stage, exhibiting elastic deformation. Then, the specimen underwent elastic–plastic deformation, and the stress–strain curve exhibited nonlinear characteristics. After peak stress, the stress–strain curve shows a hump curve, indicating a strain-softening type. The EPS beads replaced some solidified soil, reducing the volume of solidified soil per unit and weakening the bonding of soil particles. Therefore, with an increase in EPS bead content, the UCS of lightweight soil decreases.

### 3.4. Effect of Total Curing Agent Content on Stress–Strain Curves

Four groups of specimens were selected, featuring a 0.5% EPS bead content, a 15% proportion of cement replaced with fly ash, and 10%, 15%, 20%, and 25% total curing agent content; the corresponding unconfined stress–strain curves are shown in Figure 8. As the total curing agent content increased, the peak stress of the specimen increased, destructive strain decreased, and the UCS of lightweight soil increased. When the total curing agent content increased from 10% to 25%, the UCS of lightweight soil increased from 1324 kPa to 2715 kPa, and strength increased by 105%. The hydration products of the curing agent closely connect EPS beads with the soil body and transform the microstructure of the soil body into an extremely strong reticular cementation structure. Therefore, the strength of lightweight soil increases with the increase in total curing agent content.

### 3.5. Effect of Fly Ash on Stress–Strain Curves

Five groups of specimens were selected, featuring a 1% EPS bead content, a 15% total curing agent content, and 0%, 15%, 30%, 45%, and 60% proportions of cement replaced with fly ash; the corresponding unconfined stress–strain curves are shown in Figure 9. With the increase in fly ash content, the peak stress of the specimen decreases, destructive strain increases, and the UCS of lightweight soil decreases. When fly ash content is 0%, the UCS of lightweight soil is 1615 kPa; when the fly ash content is 15%, 30%, 45%, and 60%, the UCS of lightweight soil is 1518 kPa, 1412 kPa, 1265 kPa, and 1112 kPa, respectively, decreasing by 6.0%, 12.6%, 21.7%, and 31.1%. Compared to fly ash, the hydration rate of cement is faster, and there are more hydration products. Therefore, replacing the same mass of cement with fly ash reduces the strength of lightweight soil.

### 3.6. Effect of EPS Content on Unconfined Compressive Strength

The addition of lightweight materials and curing agents makes the properties of lightweight soil different from those of raw soil, but similar to those of porous cement soil. The true innovative quality of this geotechnical material is its content of lightweight material that creates hollow structures in soil, significantly reducing the soil’s weight while still providing a certain level of strength. Sixteen groups of specimens were selected, featuring a 0% proportion of cement replaced with fly ash, 10%, 15%, 20%, and 25% total curing agent content, and 0.5%, 1%, 1.5%, and 2% EPS bead content; the corresponding UCS is shown in Figure 10.

The UCS of lightweight soil decreases nonlinearly with the increase in EPS bead content. When the total curing agent content was 25%, with an increase in the EPS bead content from 0.5% to 1%, the UCS decreased from 2935 kPa to 2478 kPa, a decrease of 15.6%, and after the EPS bead content increased to 2%, the strength decreased to 1546 kPa, a decrease of 47.3%.

Therefore, the EPS bead content cannot be increased indefinitely. Although EPS beads can reduce the density of lightweight soil, excessive EPS bead addition will lead to a reduction in strength, while the solidification of soil will be lower, the solidification effect will be weaker, and sample-making will be more difficult. After fitting the data in Figure 10, it was found that when the total curing agent content and proportion of cement replaced with fly ash are unchanged, the UCS of lightweight soil and EPS bead content show an exponential function relationship, and the correlation coefficient $R^{2}$ is greater than 0.95, which satisfies the relational equation between UCS and EPS content:(1)$$q_{u}=q_{0}e^{-ta_{e}}$$

In Formula (1), $q_{u}$ is the UCS, $a_{e}$ is the EPS content, and $q_{0},t$ is a parameter between strength and EPS content, which is related to the total curing agent content. The parameters of the four curves are shown in Table 4.

### 3.7. Effect of Total Curing Agent Content on Unconfined Compressive Strength

Sixteen groups of specimens were selected, with a 15% proportion of cement replaced with fly ash, 0.5%, 1%, 1.5%, and 2% EPS bead content, and 10%, 15%, 20%, and 25% total curing agent content; the corresponding UCS is shown in Figure 11. The UCS of lightweight soil increases approximately linearly with the increase in total curing agent content. When EPS bead content was 1% and the total curing agent content increased from 10% to 25%, the UCS increased from 1121 kPa to 2317 kPa, and strength increased by 107%. When the EPS bead content was 2% and the total curing agent content increased from 10% to 25%, the UCS increased by 112%.

The curing agent plays an important role in lightweight soil. Although EPS beads can reduce the density of this type of soil, they can also decrease its strength and integrity. Mixing fly ash and cement as a curing agent can significantly improve the strength of lightweight soil, effectively compensating for the adverse effects of EPS beads on its strength and even increasing its strength, giving it the characteristics of lightness and high strength.

After fitting the data in Figure 11, it was found that when the EPS bead content and proportion of cement replaced with fly ash are unchanged, the UCS of lightweight soil and the total curing agent content exhibit an approximately linear relationship, and the correlation coefficient, $R^{2}$, is greater than 0.97. The EPS bead content only affects the slope and intercept of this linear relationship, and the relationship between UCS and total curing agent content satisfies the following relational equation:(2)$$q_{u}=ka_{c}+b$$

In Formula (2), $q_{u}$ is the UCS, $a_{c}$ is the total curing agent content, and k,b are parameters of strength and the total curing agent content, which are related to EPS bead content. The fitting parameters of the four curves are shown in Table 5.

### 3.8. Effect of Fly Ash on Unconfined Compressive Strength

Sixteen groups of specimens were selected, with a 15% total curing agent content; 0.5%, 1%, 1.5%, and 2% EPS bead content, and 0%, 15%, 30%, 45%, and 60% proportions of cement replaced with fly ash; the corresponding UCS is shown in Figure 12. The UCS of lightweight soil decreases linearly with the increase in the proportion of cement replaced with fly ash. When the EPS bead content was 1.5% and the proportion of cement replaced with fly ash was 0%, the UCS of lightweight soil was 1395 kPa; when the proportion of cement replaced with fly ash increased to 15%, 30%, 45%, and 60%, the UCS decreased to 1310 kPa, 1216 kPa, 1104 kPa, and 977 kPa, respectively, corresponding to decreases of 6.1%, 12.8%, 20.9%, and 30.0%. Although replacing cement with fly ash as a curing agent will reduce the strength of lightweight soil, as long as the proportion of cement replaced with fly ash is not too large, the reduction in the strength of lightweight soil will be less, and the corresponding performance will still meet the needs of most projects. At the same time, this technique can also consume fly ash and reduce environmental pollution. EPS lightweight soil mixed with fly ash still has advantages, and it is recommended that the proportion of cement to be replaced with fly ash should be less than 30%.

After fitting the data in Figure 12, it was found that when the EPS content and total curing agent content are unchanged, the UCS of lightweight soil and proportion of cement replaced with fly ash exhibit an approximately linear relationship, and the correlation coefficient, $R^{2}$, is greater than 0.97. The EPS content only affects the intercept and slope of the linear relationship, and the relationship between UCS and the proportion of cement replaced with fly ash satisfies the following relational equation:(3)$$q_{u}=ma_{r}+n$$

In Formula (3), $q_{u}$ is the UCS, $a_{r}$ is the proportion of cement replaced with fly ash, and m,n are the parameters relating to strength and the proportion of cement replaced with fly ash, which are related to EPS content. The fitting parameters of the four curves are shown in Table 6.

### 3.9. Model for the UCS Value of Lightweight Soil

In this experiment, the UCS of lightweight soil with different material ratios was fitted using the EPS content, the total curing agent content, and the proportion of cement replaced with fly ash as independent variables and the UCS as a dependent variable. A model formula for the relationship between the UCS of lightweight soil and EPS content, total curing agent content, and the proportion of cement replaced with fly ash was obtained.
(4)$$q_{u}=616.67-215.02e+110.83c-5.81f-27.46ec+3.3ef-0.39cf$$

In Equation (4), $q_{u}$ is the UCS, *e* is the EPS content, *c* is the total curing agent content, and *f* is the proportion of cement replaced with fly ash. The correlation coefficient $R^{2}$ exceeds 0.98. We calculated the UCS of lightweight soil with different material ratios using Equation (4). As shown in Figure 13, the error between simulated and measured values is relatively small.

### 3.10. Unconfined Compression Failure Patterns

The failure pattern of lightweight soil reflects its structural characteristics, to a certain extent. After the UCS test, the crushed specimens were photographed and recorded, and a part of the representative specimens was selected for analysis. Figure 14a shows the failure pattern of the specimen with a 0.5% EPS bead content, a 15% total curing agent content, and a 30% proportion of cement replaced with fly ash. At this point, due to the specimen’s low EPS bead content, it has fewer internal pores, and vertical cracks from top to bottom were generated after compression failure. The failure pattern is splitting failure, with no obvious signs before the failure, and the characteristics of brittle failure are significant.

Figure 14b shows the failure pattern of the specimen with a 1.5% EPS bead content, a 20% total curing agent content, and a 30% proportion of cement replaced with fly ash. The EPS bead content in this specimen is higher, but due to the larger curing agent content, the hydration products of the curing agent fill parts of the pores, so the specimen’s integration is still good and its strength is higher. After compression failure, a main crack running through the inclined surface formed, with a fracture surface at an angle of about 45°. There are small irregular cracks in other places, and the failure pattern of the specimen is oblique shear failure.

Figure 14c shows the failure pattern of the specimen with a 2% EPS bead content, 10% total curing agent content, and a 30% proportion of cement replaced with fly ash. Due to this specimen’s higher EPS bead content and lower curing agent content, it has more internal pores, and multiple wider cracks were produced after the compression failure, which finally bulged near the rupture surface. The failure pattern is bulging failure, with obvious signs before the failure, and the characteristic of plastic failure is remarkable. Through observation and analysis, under external load, stress concentration first occurs at the interface between the pores inside the specimen and EPS beads, and cracks run through these parts. EPS beads themselves will not undergo shear failure, but they will undergo a certain degree of deformation.

In summary, the failure patterns of lightweight soil mainly include splitting failure, oblique shear failure, and bulging failure, which are related to the material mix ratio. When the EPS bead content was lower, the failure pattern of the specimen was generally splitting failure, and the specimen showed obvious brittle failure. When EPS bead content and curing agent content were higher, the failure pattern of the specimen was generally oblique shear failure. When the EPS bead content was higher and the curing agent content was lower, the failure pattern of the specimen was generally bulging failure, and the specimen showed obvious plastic failure.

### 3.11. Micro Mechanisms

The physical and mechanical properties of soil are essentially related to its internal structure, and its complex properties are manifestations of micro mechanisms. The macroscopic mechanical mechanisms of lightweight soil can be understood through the study of its micro aspects.

Lightweight soil samples with different material ratios were selected for an electron microscopy scanning test. Figure 15a shows a microscopic picture of the combination of EPS beads and cured soil after being magnified 100 times, specifically the specimen with a 1.5% EPS bead content, a 10% total curing agent content, and a 15% proportion of cement replaced with fly ash. The EPS beads were wrapped in cured soil. The surface of the cured soil has obvious pores and cracks, and the pores and cracks are mainly concentrated at the EPS beads–cured-soil interface. EPS beads have a hollow honeycomb structure, and due to their larger volume, they will replace the same volume of cured soil, making pores increase in lightweight soil, so EPS beads will reduce this soil’s strength. The EPS beads’ stiffness was low, and they experienced a great deal of destructive strain. The cured soil’s stiffness was high, and it experienced little destructive strain. It is difficult to coordinate deformation. With a certain degree of deformation, the EPS beads and the cured soil can be easily separated along the interface.

Figure 15b shows a microscopic picture of the surface of the cured soil after 5000-times magnification for the specimen with a 0.5% EPS bead content, a 15% total curing agent content, and a 0% proportion of cement replaced with fly ash. The curing agent hydrated and generated needle hydrates, which filled in the pores between the EPS beads and soil and tightly connected EPS beads and soil, forming a reticular cementation structure in this space. The structural units have a more agglomerated structure, generating a certain skeleton-filling effect that increases the density of the soil structure. The curing agent plays the role of a colloid in soil, causing the bonding between soil particles to transition from contact bonding to cementation bonding, which enhances the bonding strength of the soil particles. Therefore, the curing agent increases the strength of lightweight soil. It effectively compensates for the adverse effects of EPS beads on soil strength.

Figure 16a–c show microscopic pictures of the cured soil surface after being magnified 500 times for three groups of specimens with a 1% EPS bead content, a 20% total curing agent content, and 0%, 30%, and 60% proportions of cement replaced with fly ash, respectively. The hydration products of the curing agent adhere to the surface of the soil and cover it up. When the proportion of cement replaced with fly ash is 0%, the solidified soil is relatively dense, with few cracks and pores; when the proportion of cement replaced with fly ash is 30%, fine pores and cracks can be seen on the surface of the cured soil; and when the proportion of cement replaced with fly ash is 60%, there are obvious pores and cracks on the surface of the cured soil. Replacing cement with fly ash decreases the number of hydration products and the strength of lightweight soil.

## 4. Conclusions

1. The density of EPS lightweight soil mixed with fly ash is mainly affected by EPS bead content. In total, 90% of the interior of an EPS bead is air, so, with an increase in EPS bead content, the density of lightweight soil decreases, and the effect of the curing agent on the density of lightweight soil is not significant.

2. The stress–strain curve pattern of lightweight soil is related to EPS bead content, total curing agent content, and the proportion of cement replaced with fly ash. The hydration products of the curing agent tightly connect EPS beads with soil, forming a network-bonding structure; when the total amount of curing agent increases, the strength of lightweight soil increases; when the EPS bead content increases, the volume of solidified soil decreases, the network bonding structure weakens, and the strength of lightweight soil decreases; and when the proportion of cement replaced with fly ash increases, the number of hydration products and the strength of lightweight soil decrease.

3. The failure patterns of lightweight soil mainly include splitting failure, oblique shear failure, and bulging failure, which are related to the material mix ratio. Under the action of load, when the specimen was subjected to compression failure, cracks passed through the joint surface between the EPS beads and solidified soil. When the EPS bead content was higher and the total curing agent content was lower, more cracks were generated, making the specimens prone to bulging failure near the fracture surface.

4. The reticular cementation structure formed by the hydration reaction involving the curing agent is the main source of lightweight soil’s strength, which directly affects its mechanical properties. Replacing partial cement with fly ash reduces the number of hydration products and the strength of lightweight soil. When the proportion of cement replaced with fly ash is not too high, the strength of lightweight soil decreases less, and its performance can still meet engineering needs. At the same time, this technique can also consume fly ash and reduce environmental pollution. EPS lightweight soil mixed with fly ash offers advantages, and we recommend keeping the proportion of fly ash used to replace cement less than 30%.

## Figures and Tables

**Figure 1 polymers-16-03517-f001:**
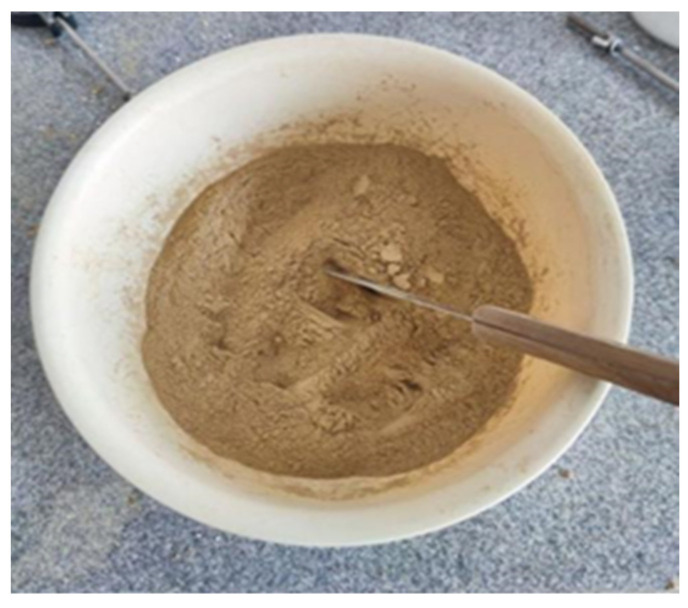
Raw soil.

**Figure 2 polymers-16-03517-f002:**
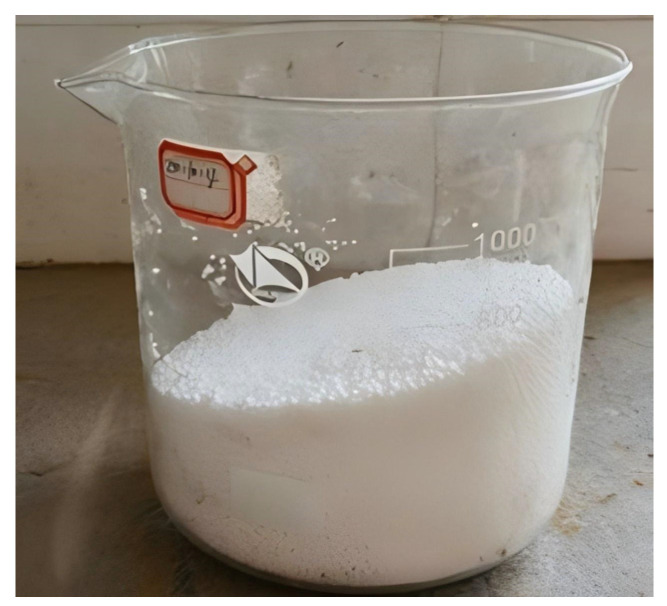
EPS beads.

**Figure 3 polymers-16-03517-f003:**
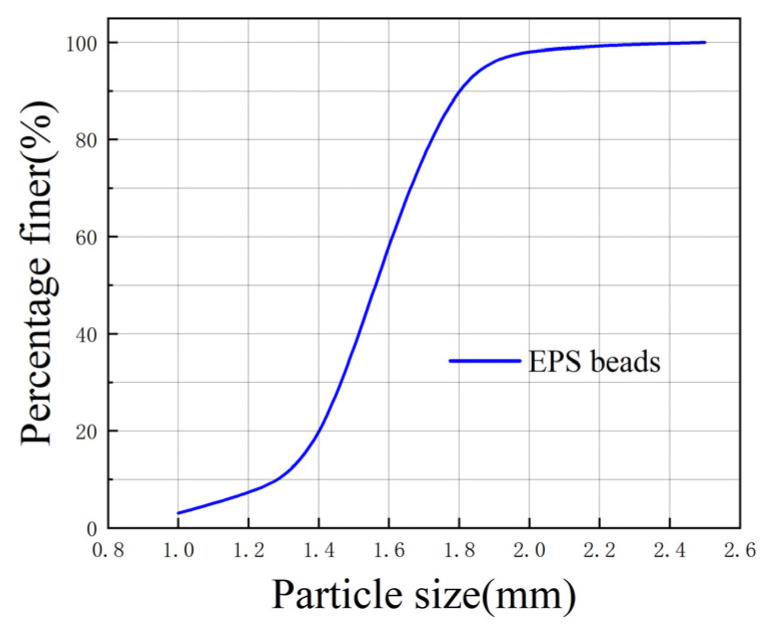
The grain-size curve of EPS beads.

**Figure 4 polymers-16-03517-f004:**
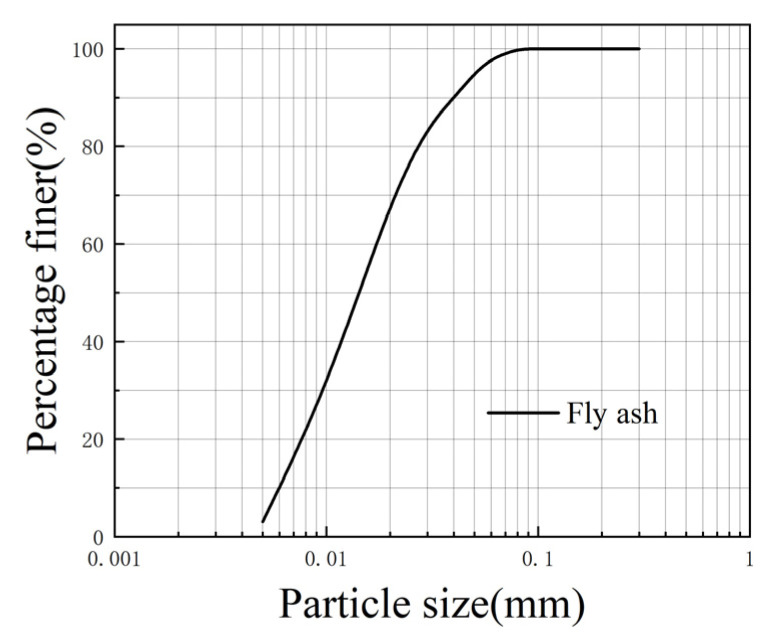
The grain-size curve of fly ash.

**Figure 5 polymers-16-03517-f005:**
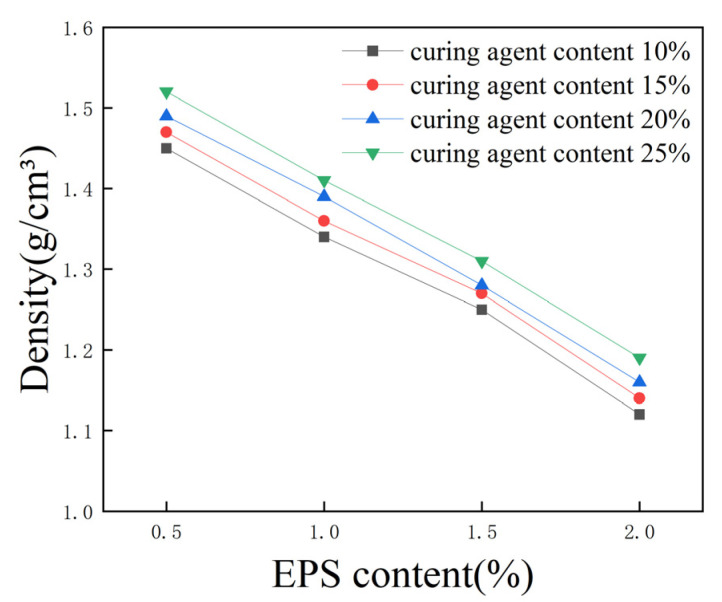
Effect of EPS content on the density of lightweight soil.

**Figure 6 polymers-16-03517-f006:**
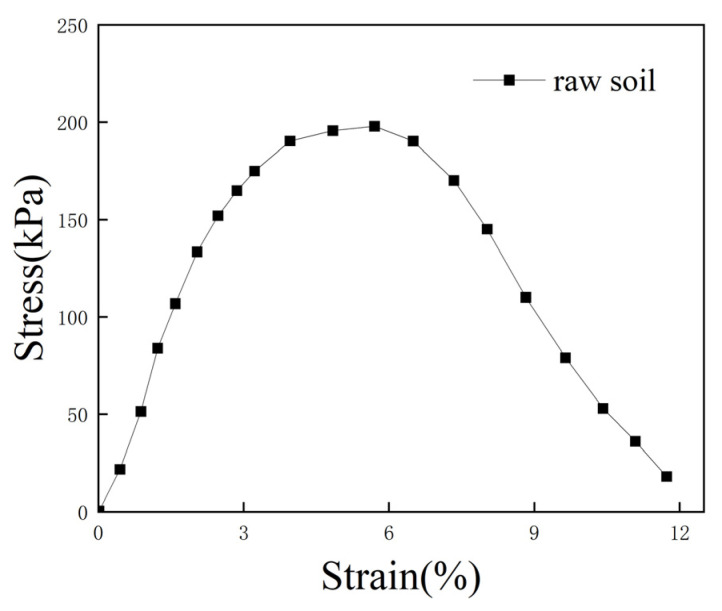
Unconfined stress–strain curve for raw soil.

**Figure 7 polymers-16-03517-f007:**
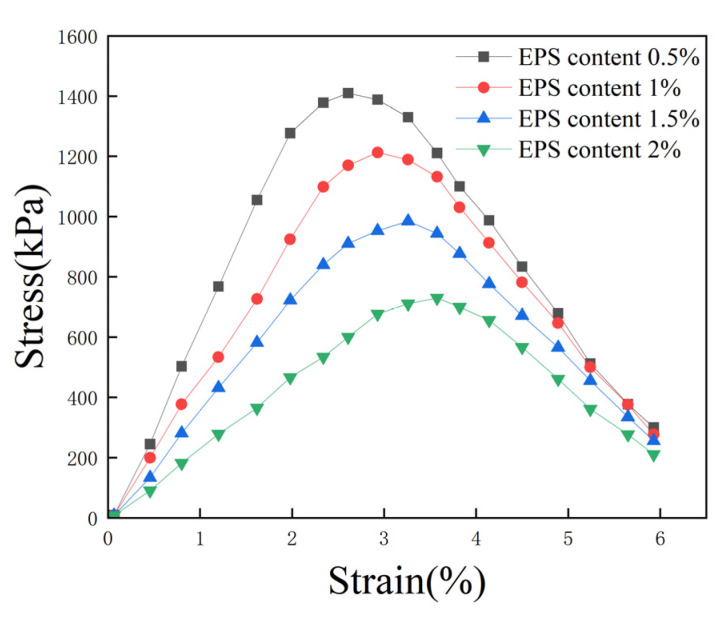
Effect of EPS content on stress–strain curve.

**Figure 8 polymers-16-03517-f008:**
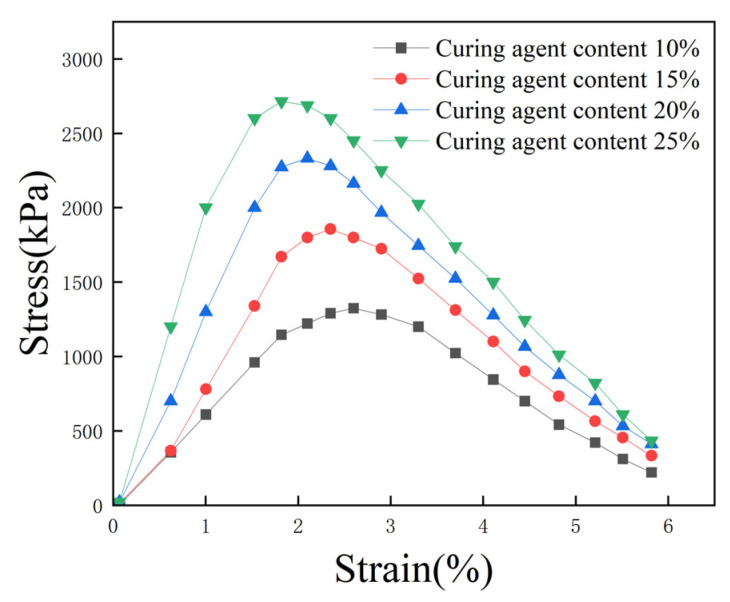
Effect of total curing agent content on stress–strain curves.

**Figure 9 polymers-16-03517-f009:**
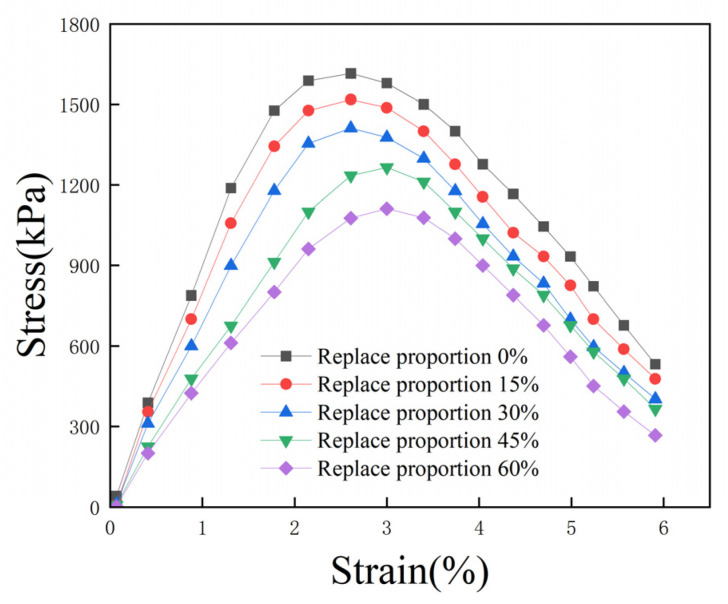
Effect of fly ash on stress–strain curves.

**Figure 10 polymers-16-03517-f010:**
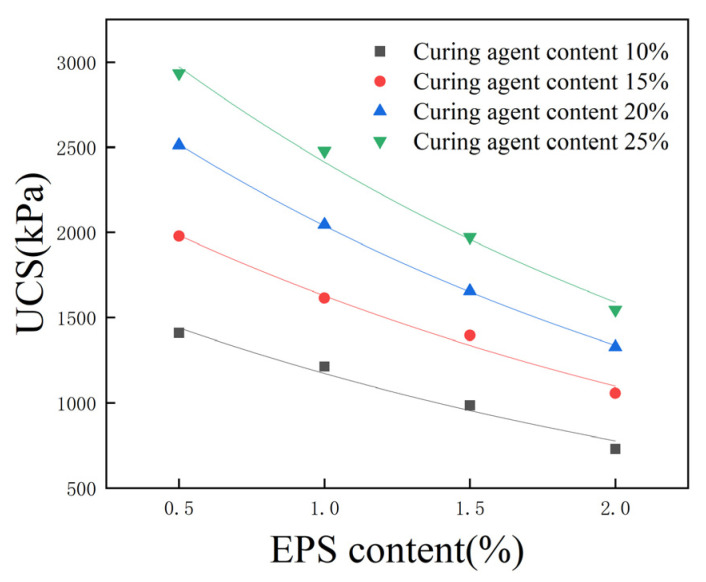
Effect of EPS content on unconfined compressive strength.

**Figure 11 polymers-16-03517-f011:**
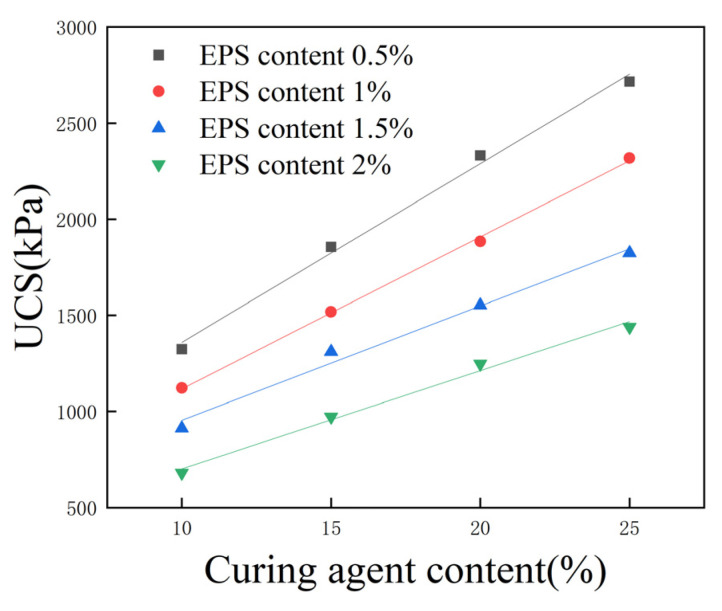
Effect of total curing agent content on unconfined compressive strength.

**Figure 12 polymers-16-03517-f012:**
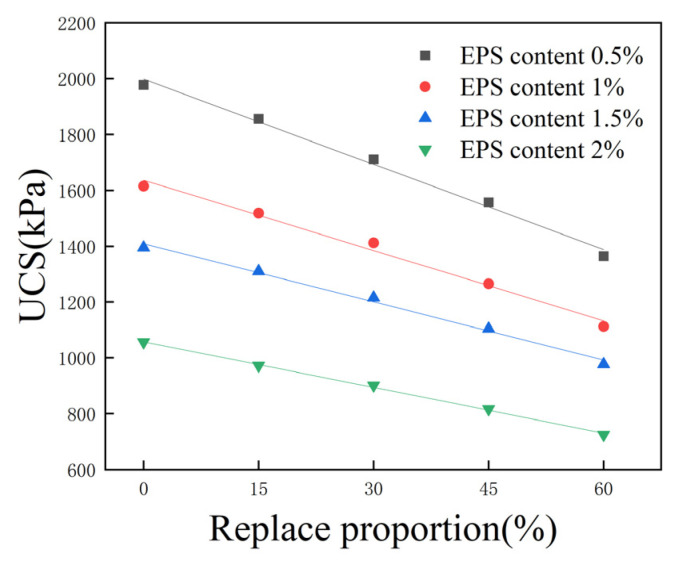
Effect of fly ash on unconfined compressive strength.

**Figure 13 polymers-16-03517-f013:**
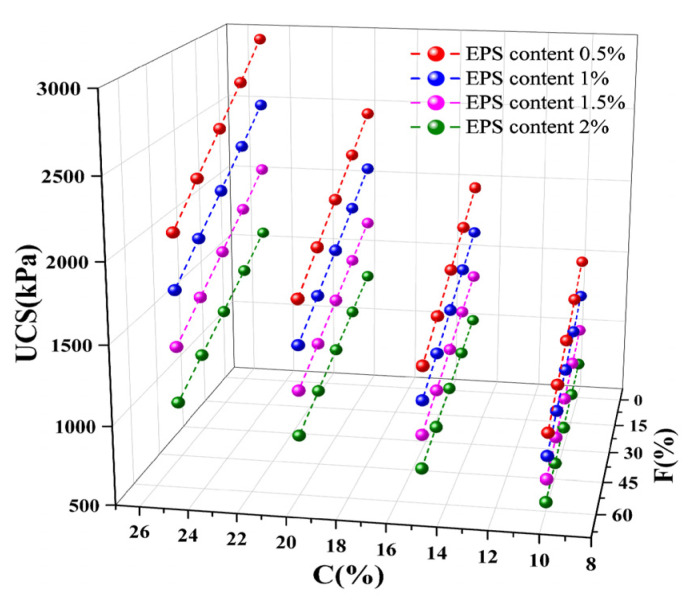
Unconfined compressive-strength simulation value.

**Figure 14 polymers-16-03517-f014:**
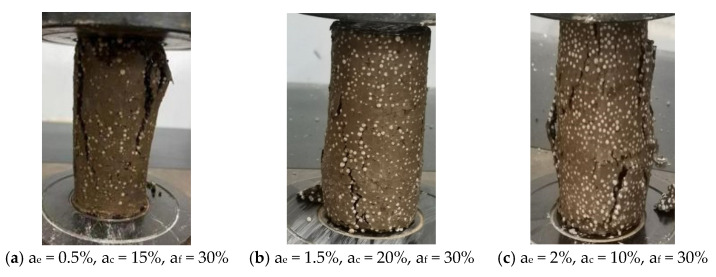
Failure patterns of lightweight soil with different material ratios.

**Figure 15 polymers-16-03517-f015:**
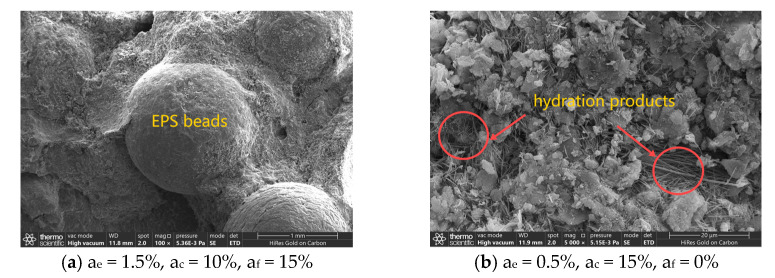
Microscopic pictures of lightweight soil at different magnifications.

**Figure 16 polymers-16-03517-f016:**
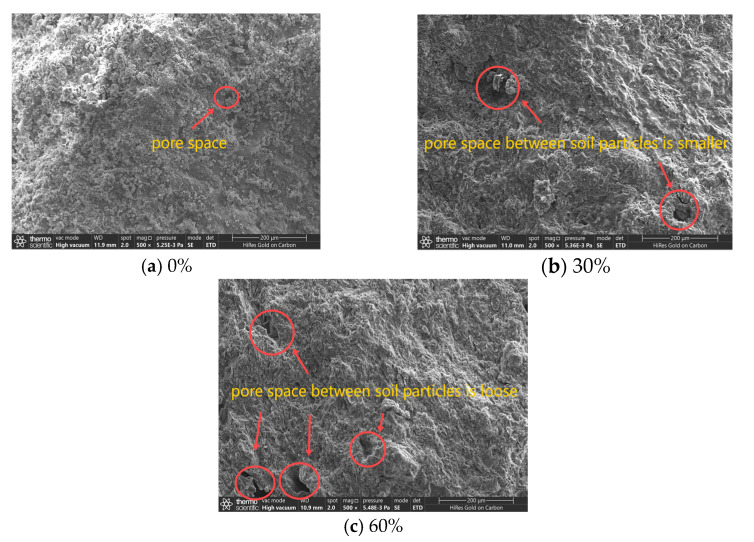
Microscopic pictures of lightweight soil with different proportions of cement replaced with fly ash.

**Table 1 polymers-16-03517-t001:** Physical parameters of raw soil.

Natural Water Content (%)	Optimum Water Content (%)	Natural Density (g/cm^3^)	Maximum Dry Density (g/cm^3^)	Plastic Limit (%)	Liquid Limit (%)
24.8	16.2	1.99	1.804	17.2	31.1

**Table 2 polymers-16-03517-t002:** Chemical compositions of cement and fly ash.

Composition	CaO	Fe_2_O_3_	SiO_2_	Al_2_O_3_	MgO	SO_3_	TiO_2_ + Na_2_O + K_2_O	Loss
cement	60.2%	3.3%	22.6%	6.3%	2.3%	1.7%	1.1%	2.5%
fly ash	6.4%	4.5%	51%	28%	0.95%	0.8%	3.6%	4.75%

**Table 3 polymers-16-03517-t003:** The test scheme for lightweight soil.

Scheme	EPS Content (%)	Curing Agent Content (%)	Replacement Proportion (%)
1	0.5, 1, 1.5, 2	10, 15, 20, 25	0
2	0.5, 1, 1.5, 2	10, 15, 20, 25	15
3	0.5, 1, 1.5, 2	15	0, 15, 30, 45, 60

**Table 4 polymers-16-03517-t004:** Fitting parameters of UCS and EPS content.

Total Curing Agent Content	$q_{0}$	t
10%	1768.17	0.41
15%	2415.61	0.39
20%	3108.61	0.42
25%	3662.59	0.42

**Table 5 polymers-16-03517-t005:** Fitting parameters of UCS and total curing agent content.

EPS Content	k	b
0.5%	92.96	429.7
1%	79.1	326
1.5%	59.5	358.5
2%	51.1	190

**Table 6 polymers-16-03517-t006:** Fitting parameters regarding lightweight soil strength and the proportion of cement replaced with fly ash.

EPS Content	m	n
0.5%	−10.2	1998
1%	−8.4	1636.2
1.5%	−6.9	1408.8
2%	−5.5	1057.6

## Data Availability

The original contributions presented in this study are included in the article/Appendix A. Further inquiries can be directed to the corresponding author or https://doi.org/10.6084/m9.figshare.28040171.

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
