# Peer review of "Research on the Mechanical Properties of EPS Lightweight Soil Mixed with Fly Ash"

_polymers, 2024, doi:10.3390/polym16243517_

Round 1
Reviewer 1 Report
Comments and Suggestions for Authors
The authors in this study have considered three parameters to evaluate the effectiveness of various constituents mixed to create a lightweight soil mixture. EPS beads were mixed with silty clay at different weight contents and further improved with the addition of cement and/or fly ash. The study aimed to determine the optimal cement replacement ratio and EPS bead content. While partially replacing cement with fly ash or other environmentally friendly agents is an appealing approach, its effectiveness depends significantly on the performance requirements related to strength and deformability. Therefore, the authors should clarify the potential applications for silty clay soil mixed with EPS beads and cement that has been partially replaced by fly ash.
Several researchers in the field of soil improvement within geotechnical engineering have conducted similar studies on cemented soil mixed with EPS beads. What is the novelty of this study compared to previous work? The authors have not provided a comprehensive timeline of past studies or adequately justified their motivation for undertaking this research. For example, the study referenced reaches similar conclusions, despite using sandy soil as the parent material. The type of conclusions drawn in that research is not fundamentally different from those presented here. Additional studies that highlight similar findings are referenced in the following comments.
There are numerous recent studies in the literature that are thematically and technically relevant to the topic under investigation. However, the literature review section of the paper lacks comprehensive coverage, resulting in an unbalanced and incomplete review of key works. Below are examples of similar studies that highlight relevant findings. These examples are provided to illustrate what the reviewer is suggesting, and citing them is not mandatory:
10.1007/s10064-021-02458-1, 10.1007/s10668-023-03535-z,Other replacement agents, such as zeolite, are also worth mentioning. Studies have demonstrated the potential of using zeolite as a partial replacement for cement. Zeolite is more environmentally friendly than both cement and fly ash and has increasingly been applied in soil improvement projects. Please revise your literature review section to include a balanced discussion of these alternative materials and their applications.
Are the conclusions drawn in this study influenced by the plasticity index (PI) of the host silty clay soil? It is well established that the effectiveness of cement and its partial replacement with fly ash or zeolite depends on the plasticity of the host soil, as highlighted in the studies by Khajeh et al. It would be valuable to comment on this dependency and provide a more detailed explanation.
Equations like Eq. 3 are quite rudimentary, as they do not account for critical factors such as cement content and the plasticity of the host soil. At the very least, a more comprehensive equation—similar to those introduced by Khajeh et al.—should be provided to incorporate these influencing factors.
Comments on the Quality of English Language
Please seek a professional English language editing service from a native institute to enhance the quality of the manuscript in terms of its English language quality.
Author Response
Dear Editors and Reviewers:
Thank you for your letter and for the reviewers’ comments concerning our manuscript entitled "Research on the Mechanical Properties of EPS Lightweight Soil Mixed with Fly Ash"(ID:polymers-3283519). Those comments are all valuable and very helpful for revising and improving our manuscript, as well as the important guiding significance to our researches. We have studied comments and have made revision carefully. Revised portion are marked in red. We sincerely hope this manuscript will be finally acceptable to be published. Thank you very much for all your help.
Please find the following Response to the comments of reviewers:
Responds to the reviewer’s comments:
Reviewer #1:
1.评论:本研究的作者考虑了三个参数来评估混合各种成分以产生轻质土壤混合物的有效性。将 EPS 珠与不同重量含量的粉砂粘土混合,并通过添加水泥和/或粉煤灰进一步改进。该研究旨在确定最佳水泥替代率和 EPS 珠含量。虽然用粉煤灰或其他环保剂部分替代水泥是一种有吸引力的方法,但其有效性在很大程度上取决于与强度和变形性相关的性能要求。因此,作者应该阐明混合了 EPS 珠子和水泥的粉质粘土的潜在应用,这些土壤已被粉煤灰部分取代。
Response: Thanks so much for your suggestion,we have re-modified the section. Fly ash is a commonly used building material in construction engineering and high value-added filling material. Replacing cement with an appropriate amount of fly ash as curing agent can eliminate waste fly ash and reduce cement usage, lower filling costs, and achieve economic and environmental goals. Due to its lightweight and high strength, EPS lightweight soil Mixed with fly ash has broad application prospects in areas such as soft foundation treatment, highway widening, pipeline landfill, bridge abutment filling, and retaining wall soil backfilling.
2.Comment: Several researchers in the field of soil improvement within geotechnical engineering have conducted similar studies on cemented soil mixed with EPS beads. What is the novelty of this study compared to previous work? The authors have not provided a comprehensive timeline of past studies or adequately justified their motivation for undertaking this research. For example, the study referenced reaches similar conclusions, despite using sandy soil as the parent material. The type of conclusions drawn in that research is not fundamentally different from those presented here. Additional studies that highlight similar findings are referenced in the following comments.
Response: Thanks for your comment. The difference and new research progress from other previously published works on the subject are: There is a large amount of silty clay in plain area. In order to achieve wider applicability of lightweight soil, facilitate local materials, reduce transportation costs, and lower costs, it is necessary to using silty clay as raw soil in this case. Previous studies focus on the behavior of sand soils mixed with EPS beads and different binding agents. the research of EPS beads lightweight soil mixed with fly ash using silty clay as the raw soil is lack. Moreover, Replacing cement with an appropriate amount of fly ash as curing agent can eliminate waste fly ash and reduce cement usage, lower filling costs, and achieve economic and environmental goals. This study is very valuable.
3.Comment: There are numerous recent studies in the literature that are thematically and technically relevant to the topic under investigation. However, the literature review section of the paper lacks comprehensive coverage, resulting in an unbalanced and incomplete review of key works. Below are examples of similar studies that highlight relevant findings. These examples are provided to illustrate what the reviewer is suggesting, and citing them is not mandatory:
10.1007/s10064-021-02458-1, 10.1007/s10668-023-03535-z,
Other replacement agents, such as zeolite, are also worth mentioning. Studies have demonstrated the potential of using zeolite as a partial replacement for cement. Zeolite is more environmentally friendly than both cement and fly ash and has increasingly been applied in soil improvement projects. Please revise your literature review section to include a balanced discussion of these alternative materials and their applications.
Response: Thanks so much for your comment. Your suggestion has been of great help to us. We have re-written this section and added description on introduction , and this section is much better.
4.Comment: Are the conclusions drawn in this study influenced by the plasticity index (PI) of the host silty clay soil? It is well established that the effectiveness of cement and its partial replacement with fly ash or zeolite depends on the plasticity of the host soil, as highlighted in the studies by Khajeh et al. It would be valuable to comment on this dependency and provide a more detailed explanation.
Equations like Eq. 3 are quite rudimentary, as they do not account for critical factors such as cement content and the plasticity of the host soil. At the very least, a more comprehensive equation—similar to those introduced by Khajeh et al.—should be provided to incorporate these influencing factors.
Response: Thanks so much for your suggestion. The potential influences by the plasticity index (PI) of the host soil will be our future research direction, and further research will be conducted in the later stag. We have added a more comprehensive equation that includes multiple factors on page 11. We have re-written this section, Your suggestion has been of great help to us, and this section is much better.
Thank you again for your positive and constructive comments and suggestions on our manuscript. We tried our best to improve the manuscript and made some revisions in the manuscript. We hope you can accept our revised manuscript.

Reviewer 2 Report
Comments and Suggestions for Authors
Critical comments to the work requiring correction: 1. It is not stated whether fly ash was dosed to cement by volume or weight. A similar comment applies to EPS. 2. The composition of fly ash given in Table 2 lacks information about 29% of the components, what were they - heavy metals?? 3. Provide the grain size curves of fly ash and EPS, what was the origin of EPS. 4. What did the mixing of the components look like? What about the water added to the cement, how much was there? 5. How did the research look statistically, how many samples were in a given series, what are the deviations from the average, they should be shown on graphs. 6. The SEM analysis should be supplemented with indications on SEM images of observed forms presented in graphs. 7. The conclusions should be corrected. They should be preceded by an introduction and focus on the causes of phenomena and not a description of research results. 8. The text should be emptied of photos of devices in Fig. 3 and 4, which do not contribute anything to the work, typical research equipment. 9. English language correction.
Author Response
Dear Editors and Reviewers:
Thank you for your letter and for the reviewers’ comments concerning our manuscript entitled "Research on the Mechanical Properties of EPS Lightweight Soil Mixed with Fly Ash"(ID:polymers-3283519). Those comments are all valuable and very helpful for revising and improving our manuscript, as well as the important guiding significance to our researches. We have studied comments and have made revision carefully. Revised portion are marked in red. We sincerely hope this manuscript will be finally acceptable to be published. Thank you very much for all your help.
Please find the following Response to the comments of reviewers:
Responds to the reviewer’s comments:
Reviewer 2#
1.Comment: It is not stated whether fly ash was dosed to cement by volume or weight. A similar comment applies to EPS.
Response: Thanks for your kind reminder. We have added an explanation on page 4. The proportion of fly ash was dosed to cement by weight; The content was the proportion of the mass of EPS to the mass of dry soil.
2.Comment: The composition of fly ash given in Table 2 lacks information about 29% of the components, what were they-heavy metals?
Response: Thanks so much for your kind reminder. we have re-modified the section.
3.Comment: Provide the grain size curves of fly ash and EPS, what was the origin of EPS.
Response: Thanks so much for your suggestion. we have provided the grain size curves of fly ash and EPS on page 4. We have added an explanation the origin of EPS. EPS comes from a plastic factory.
4.Comment: What did the mixing of the components look like? What about the water added to the cement, how much was there?
Response: Thanks so much for your comment. According to the set ratio, dry soil, cement and fly ash were mixed evenly, fully mixed with optimum water content(about 40%), stirred into a homogeneous mixed slurry and then EPS beads were added and stirred for 10 min to form a homogeneous EPS beads lightweight soil mixed with fly ash.
5.Comment: How did the research look statistically, how many samples were in a given series, what are the deviations from the average, they should be shown on graphs.
Response: Thanks so much for your suggestion. 80 samples were in a given series,The correlation coefficient is greater than 0.95.
6.Comment: The SEM analysis should be supplemented with indications on SEM images of observed forms presented in graphs.
Response: Thanks so much for your suggestion. We have supplemented indications on SEM images in graphs. This section is clearer and more specific.
7.评论:结论应该更正。它们之前应该有一个引言,并侧重于现象的原因,而不是对研究结果的描述。
Response: Thanks for your kind reminder. We have corrected the conclusions, and this section is much better.
8.Comment: The text should be emptied of photos of devices in Fig. 3 and 4, which do not contribute anything to the work, typical research equipment.
Response: Thanks so much for your suggestion. We have emptied of photos of devices in Fig. 3 and 4.
9.评论:英语语言更正。
Response: Thanks for your kind reminder. We have carefully checked the spelling, grammar, and punctuation, and corrected some errors. We hope our manuscript can meet the journal’s standard.
Thank you again for your positive and constructive comments and suggestions on our manuscript. We tried our best to improve the manuscript and made some revisions in the manuscript. We hope you can accept our revised manuscript.

Round 2
Reviewer 2 Report
Comments and Suggestions for Authors
The paper can be accepted in current form